# FishNet: A Versatile Backbone for Image, Region, and Pixel Level Prediction

**Shuyang Sun**[1], **Jiangmiao Pang**[3], **Jianping Shi**[2], **Shuai Yi**[2], **Wanli Ouyang**[1]

[1]The University of Sydney  [2]SenseTime Research  [3]Zhejiang University

shuyang.sun@sydney.edu.au

## Abstract

The basic principles in designing convolutional neural network (CNN) structures for predicting objects on different levels, e.g., image-level, region-level, and pixel-level, are diverging. Generally, network structures designed specifically for image classification are directly used as default backbone structure for other tasks including detection and segmentation, but there is seldom backbone structure designed under the consideration of unifying the advantages of networks designed for pixel-level or region-level predicting tasks, which may require very deep features with high resolution. Towards this goal, we design a fish-like network, called FishNet. In FishNet, the information of all resolutions is preserved and refined for the final task. Besides, we observe that existing works still cannot *directly* propagate the gradient information from deep layers to shallow layers. Our design can better handle this problem. Extensive experiments have been conducted to demonstrate the remarkable performance of the FishNet. In particular, on ImageNet-1k, the accuracy of FishNet is able to surpass the performance of DenseNet and ResNet with fewer parameters. FishNet was applied as one of the modules in the winning entry of the COCO Detection 2018 challenge. The code is available at https://github.com/kevin-ssy/FishNet.

## 1   Introduction

Convolutional Neural Network (CNN) has been found to be effective for learning better feature representations in the field of computer vision [17, 26, 28, 9, 37, 27, 4]. Thereby, the design of CNN becomes a fundamental task that can help to boost the performance of many other related tasks. As the CNN becomes increasingly deeper, recent works endeavor to refine or reuse the features from previous layers through identity mappings [8] or concatenation [13].

The CNNs designed for image-level, region-level, and pixel-level tasks begin to diverge in network structure. Networks for image classification use consecutive down-sampling to obtain deep features of low resolution. However, the features with low resolution are not suitable for pixel-level or even region-level tasks. Direct use of high-resolution shallow features for region and pixel-level tasks, however, does not work well. In order to obtain deeper features with high resolution, the well-known network structures for pixel-level tasks use U-Net or hourglass-like networks [22, 24, 30]. Recent works on region-level tasks like object detection also use networks with up-sampling mechanism [21, 19] so that small objects can be described by the features with relatively high resolution.

Driven by the success of using high-resolution features for region-level and pixel-level tasks, this paper proposes a fish-like network, namely FishNet, which enables the features of high resolution to contain high-level semantic information. In this way, features pre-trained from image classification are more friendly for region and pixel level tasks.

We carefully design a mechanism that have the following three advantages.

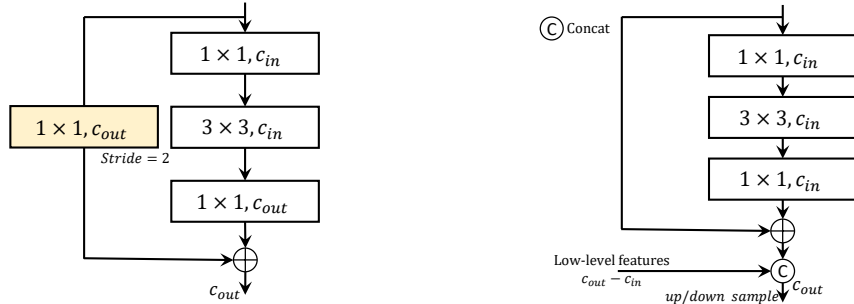

Figure 1: The up/down-sampling block for ResNet (*left*), and FishNet (*right*). The $1 \times 1$ convolution layer in yellow indicates the *Isolated convolution (I-conv, see Section 2)*, which makes the direct BP incapable and degrades the gradient from the output to shallow layers.

First, **it is the first backbone network that unifies the advantages of networks designed for pixel-level, region-level, and image-level tasks.** Compared to the networks designed purely for the image classification task, our network as a backbone is more effective for pixel-level and region-level tasks.

Second, **it enables the gradient from the very deep layer to be *directly* propagated to shallow layers, called direct BP in this paper.** Recent works show that there are two designs that can enable direct BP, identity mapping with residual block [8] and concatenation [13]. However, the untold fact is that existing network designs, e.g. [9, 8, 13, 28, 34, 32], still do not enable direct BP. This problem is caused by the convolutional layer between features of different resolutions. As shown in the Figure 1, the ResNet [9] utilize a convolutional layer with stride on the skip connection to deal with the inconsistency between the numbers of input and output channels, which makes the identity mapping inapplicable. Convolution without identity mapping or concatenation degrades the gradient from the output to shallow layers. Our design better solves this problem by concatenating features of very different depths to the final output. We also carefully design the components in the network to ensure the direct BP. With our design, the semantic meaning of features are also preserved throughout the whole network.

Third, **features of very different depth are preserved and used for refining each other.** Features with different depth have different levels of abstraction of the image. All of them should be kept to improve the diversity of features. Because of their complementarity, they can be used for refining each other. Therefore, we design a feature preserving-and-refining mechanism to achieve this goal.

A possibly counter-intuitive effect of our design is that it performs better than traditional convolutional networks in the trade-off between the number of parameters and accuracy for image classification. The reasons are as follows: 1) the features preserved and refined are complementary to each other and more useful than designing networks with more width or depth; and 2) it facilitates the direct BP. Experimental results show that our compact model FishNet-150, of which the number of parameters is close to ResNet-50, is able to surpass the accuracy of ResNet-101 and DenseNet-161(k=48) on ImageNet-1k. For region and pixel level tasks like object detection and instance segmentation, our model as a backbone for Mask R-CNN [10] improves the absolute AP by $2.8\%$ and $2.3\%$ respectively on MS COCO compared to the baseline ResNet-50.

## 1.1 Related works

**CNN architectures for image classification.** The design of deep CNN architecture is a fundamental but challenging task in deep learning. Networks with better design extract better features, which can boost the performance of many other tasks. The remarkable improvement in the image recognition challenge ILSVRC [25] achieved by AlexNet [17] symbolizes a new era of deep learning for computer vision. After that, a number of works, e.g. VGG [26], Inception [28], all propose to promote the network capability by making the network deeper. However, the network at this time still cannot be too deep because of the problem of vanishing gradient. Recently, the problem of vanishing gradient is greatly relieved by introducing the skip connections into the network [9]. There is a series of on-going works on this direction [29, 34, 32, 13, 2, 11, 31, 33]. However, among all these networks designed for image classification, the features of high resolution are extracted by the shallow layers with small receptive field, which lack the high-level semantic meaning that can only be obtained on

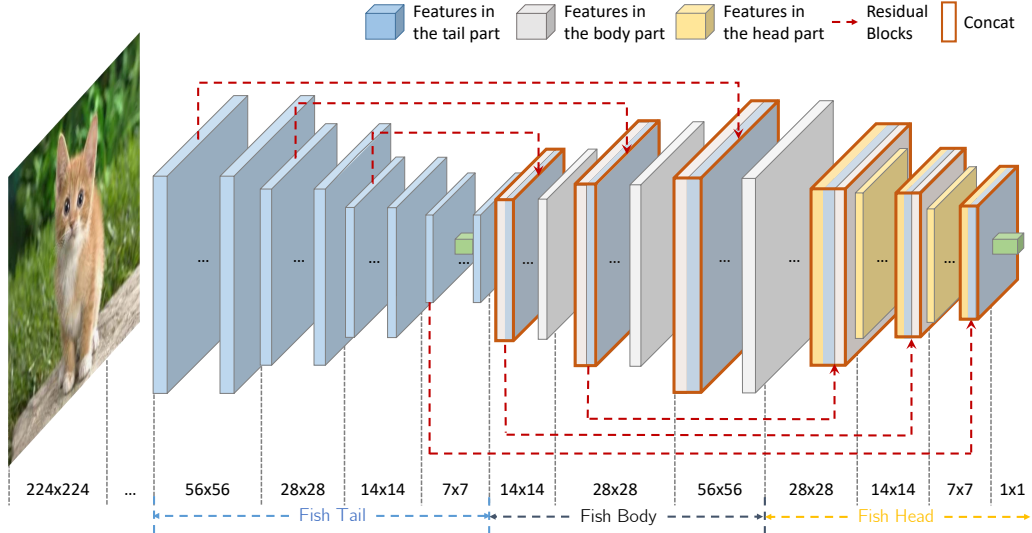

Figure 2: Overview of the FishNet. It has three parts. *Tail* uses existing works to obtain deep low-resolution features from the input image. *Body* obtains high-resolution features of high-level semantic information. *Head* preserves and refines the features from the three parts.

deeper layers. Our work is the first to extract high-resolution deep feature with high-level semantic meaning and improve image classification accuracy at the same time.

**Design in combining features from different layers.** Features from different resolution or depth could be combined using nested sparse networks [16], hyper-column [7], addition [18] and residual blocks [22, 21](conv-deconv using residual blocks). Hyper-column networks directly concatenate features from different layers for segmentation and localization in [7]. However, features from deep layers and shallow layers were not used for refining each other. Addition [18] is a fusion of the features from deep and shallow layers. However, addition only mix the features of different abstraction levels, but cannot preserve or refine both of them. Concatenation followed by convolution is similar to addition [33]. When residual blocks [22, 21], also with addition, are used for combining features, existing works have a pre-defined target to be refined. If the skip layer is for the deep features, then the shallow features serve only for refining the deep features, which will be discarded after the residual blocks in this case. In summary, addition and residual blocks in existing works do not preserve features from both shallow and deep layers, while our design preserves and refines them.

**Networks with up-sampling mechanism.** As there are many other tasks in computer vision, e.g. object detection, segmentation, that require large feature maps to keep the resolution, it is necessary to apply up-sampling methods to the network. Such mechanism often includes the communication between the features with very different depths. The series of works including U-Net [24], FPN [21], stacked hourglass [22] etc., have all shown their capability in pixel-level tasks [22] and region-level tasks [21, 19]. However, none of them has been proven to be effective for the image classification task. MSDNet [12] tries to keep the feature maps with large resolution, which is the most similar work to our architecture. However, the architecture of MSDNet still uses convolution between features of different resolutions, which cannot preserve the representations. Besides, it does not provide an up-sampling pathway to enable features with large resolution and more semantic meaning. The aim of MSDNet introducing the multi-scale mechanism into its architecture is to do budget prediction. Such design, however, did not show improvement in accuracy for image classification. Our FishNet is the first in showing that the U-Net structure can be effective for image classification. Besides, our work preserves and refines features from both shallow and deep layers for the final task, which is not achieved in existing networks with up-sampling or MSDNet.

**Message passing among features/outputs.** There are some approaches using message passing among features for segmentation [36], pose estimation [3] and object detection [35]. These designs are based on backbone networks, and the FishNet is a backbone network complementary to them.

## 2 Identity Mappings in Deep Residual Networks and Isolated Convolution

The basic building block for ResNet is called the residual block. The residual blocks with identity mapping [8] can be formulated as

$$x_{l+1} = x_l + \mathcal{F}(x_l, W_l), \tag{1}$$

where $x_l$ denotes the input feature for the residual block at layer $l$, and $\mathcal{F}(x_l, W_l)$ denotes the residual function with input $x_l$ and parameters $W_l$. We consider the stack of all residual blocks for the same resolution as a *stage*. Denote the feature at the $l$th layer of stage $s$ by $x_{l,s}$. We have:

$$x_{L_s,s} = x_{0,s} + \sum_{l=1}^{L_s} \mathcal{F}(x_{l,s}, W_{l,s}), \quad \frac{\partial \mathcal{L}}{\partial x_{0,s}} = \frac{\partial \mathcal{L}}{\partial x_{L_s,s}}(1 + \frac{\partial}{\partial x_{0,s}} \sum_{l=1}^{L_s} \mathcal{F}(x_{l,s}, W_{l,s})) \tag{2}$$

where $L_s$ denotes the number of stacked residual blocks at the stage $s$, $\mathcal{L}$ is a loss function. The additive term $\frac{\partial \mathcal{L}}{\partial x_{L_s,s}}$ in (2) ensures that the gradient of $x_{L_s,s}$ can be *directly propagated* to $x_{0,s}$. We consider features with different resolutions as having different stages. In the original ResNet, the features of different resolutions are different in number of channels. Therefore, a transition function $h(\cdot)$ is needed to change the number of channels before down-sampling:

$$x'_{0,s+1} = h(x_{L_s,s}) = \sigma(\lambda_s \otimes x_{L_s,s} + b_{L_s,s}) \tag{3}$$

where $\sigma(\cdot)$ is the activation function. $\lambda_s$ is the filter and $b_{L_s,s}$ is the bias at the transition layer of stage $s$ respectively. The symbol $\otimes$ represents the convolution. Since the numbers of channels for $x_{L_s,s}$ and $x'_{0,s+1}$ are different, identity mapping is not applicable.

**Gradient propagation problem from Isolated convolution (I-conv).** Isolated convolution (I-conv) is the convolution in (3) without identity mapping or concatenation. As analyzed and validated by experiments in [8], it is desirable to have the gradients from a deep layer directly transmitted to shallow layers. Residual blocks with identity mapping [8] and dense block with concatenation [13] facilitate such direct gradient propagation. Gradients from the deep layer cannot be directly transmitted to the shallow layers if there is an I-conv. The I-conv between features with different resolutions in ResNet [8] and the I-conv (called transition layer in [13]) between adjacent dense blocks, however, hinders the direct gradient propagation. Since ResNet and DenseNet still have I-convs, the gradients from the output cannot be directly propagated to shallow layers for them, similarly for the networks in [17, 26]. The invertible down-sampling in [15] avoids the problem of I-conv by using all features from the current stage for the next stage. The problem is that it will exponentially increase the number of parameters as the stage ID increases (188M in [15]).

We have identified the gradient propagation problem of I-conv in existing networks. Therefore, we propose a new architecture, namely FishNet, to solve this problem.

## 3 The FishNet

Figure 2 shows an overview of the FishNet. The whole "fish" is divided into three parts: tail, body, and head. The fish tail is an existing CNN, e.g. ResNet, with the resolution of features becoming smaller as the CNN goes deeper. The fish body has several up-sampling and refining blocks for refining features from the tail and the body. The fish head has several down-sampling and refining blocks for preserving and refining features from the tail, body and head. The refined features at the last convolutional layer of the head are used for the final task.

**Stage** in this paper refers to a bunch of convolutional blocks fed by the features with the same resolution . Each part in the FishNet could be divided into several stages according to the resolution of the output features. **With the resolution becoming smaller, the stage ID goes higher.** For example, the blocks with outputs resolution $56 \times 56$ and $28 \times 28$ are at stage 1 and 2 respectively in all the three parts of the FishNet. Therefore, in the fish tail and head, the stage ID is becoming higher while forwarding, while in the body part the ID is getting smaller.

Figure 3 shows the interaction among tail, body, and head for features of two stages. The fish tail in Figure 3(a) could be regarded as a residual network. The features from the tail undergo several residual blocks and are also transmitted to the body through the horizontal arrows. The body in Figure 3(a) preserves both the features from the tail and the features from the previous stage of the body by concatenation. Then these concatenated features will be up-sampled and refined with details

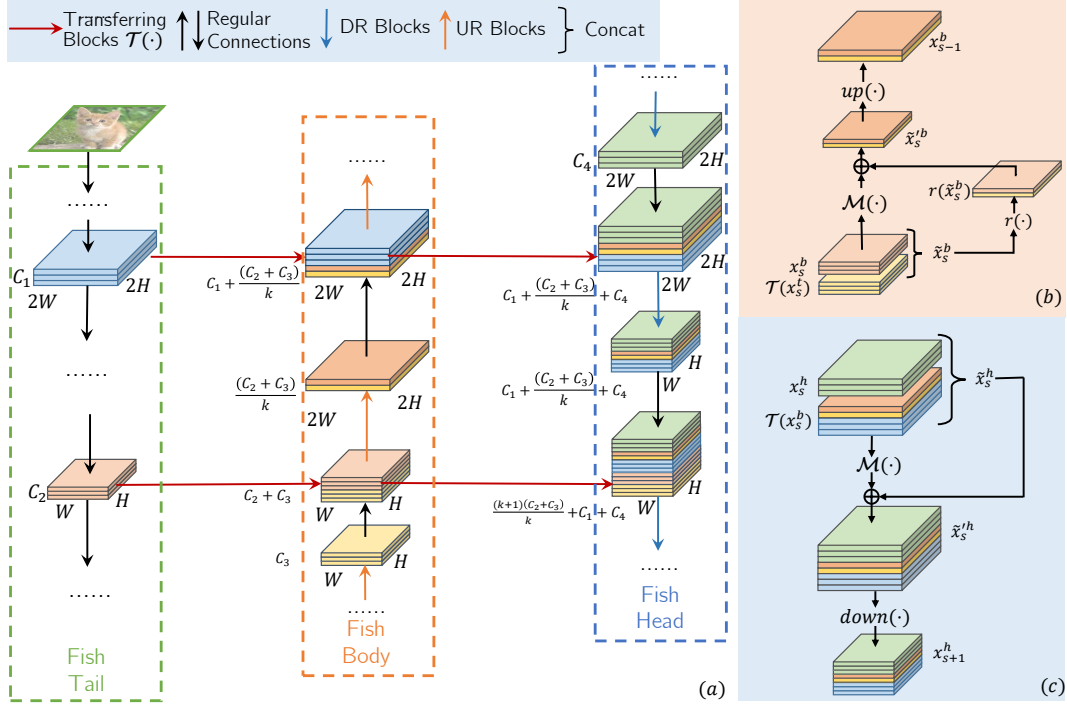

Figure 3: (Better seen in color and zoomed in.) **(a) Interaction among the tail, body and head for features of two stages,** the two figures listed on the right exhibit the detailed structure for **(b) the Up-sampling & Refinement block (UR-block),** and **(c) the Down-sampling & Refinement block (DR-block).** In the Figure (a), feature concatenation is used when vertical and horizontal arrows meet. The notations $C_*, *H, *W$ denote the number of channels, height, and width respectively. $k$ represents the channel-wise reduction rate described in Equation 8 and Section 3.1. Note that there is no Isolated convolution (I-conv) in the fish body and head. Therefore, the gradient from the loss can be directly propagated to shallow layers in tail, body and head.

shown in Figure 3(b) and the details about the UR-block will be discussed in Section 3.1. The refined features are then used for the head and the next stage of the body. The head preserves and refines all the features from the body and the previous stage of the head. The refined features are then used for the next stage of the head. Details for message passing at the head are shown in Figure 3(c) and discussed in Section 3.1. The horizontal connections represent the transferring blocks between the tail, the body and the head. In Figure 3(a), we use the residual block as the transferring blocks.

## 3.1 Feature refinement

In the FishNet, there are two kinds of blocks for up/down sampling and feature refinement: the Up-sampling & Refinement block (UR-block) and Down-sampling & Refinement block (DR-block).

**The UR-block.** Denote the output features from the first layer at the stage $s$ by $x_s^t$ and $x_s^b$ for the tail and body respectively. $s \in \{1, 2, ..., min(N^t - 1, N^b - 1)\}$, $N^t$ and $N^b$ represent the number of stages for the tail part and the body part. Denote feature concatenation as $concat(\cdot)$. The UR-block can be represented as follows:

$$x_{s-1}^b = UR(x_s^b, \mathcal{T}(x_s^t)) = up(\tilde{x}_s'^b) \tag{4}$$

where the $\mathcal{T}$ denotes residual block transferring the feature $x_{s-1}^t$ from tail to the body, the $up(\tilde{x}_s'^b)$ represents the feature refined from the previous stage in the fish body. The output $x_{s-1}^b$ for next stage

is refined from $x_s^t$ and $x_s^b$ as follows:

$$x_{s-1}^b = up(\tilde{x}_s'^b), \tag{5}$$

$$\tilde{x}_s'^b = r(\tilde{x}_s^b) + \mathcal{M}(\tilde{x}_s^b), \tag{6}$$

$$\tilde{x}_s^b = concat(x_s^b, \mathcal{T}(x_s^t)), \tag{7}$$

where $up(\cdot)$ denotes the up-sampling function. As a summary, the UR-block concatenates features from body and tail in (7) and refine them in (6), then upsample them in (5) to obtain the output $x_{s-1}^b$. The $\mathcal{M}$ in (6) denotes the function that extracts the message from features $\tilde{x}_s^b$. We implemented $\mathcal{M}$ as convolutions. Similar to the residual function $\mathcal{F}$ in (1), the $\mathcal{M}$ in (6) is implemented by bottleneck Residual Unit [8] with 3 convolutional layers. The channel-wise reduction function $r$ in (6) can be formulated as follows:

$$r(x) = \hat{x} = [\hat{x}(1), \hat{x}(2), \dots, \hat{x}(c_{out})], \quad \hat{x}(n) = \sum_{j=0}^{k} x(k \cdot n + j), n \in \{0, 1, ..., c_{out}\}, \tag{8}$$

where $x = \{x(1), x(2), \dots, x(c_{in})\}$ denotes $c_{in}$ channels of input feature maps and $\hat{x}$ denotes $c_{out}$ channels of output feature maps for the function $r$, $c_{in}/c_{out} = k$. It is an element-wise summation of feature maps from the adjacent $k$ channels to 1 channel. We use this simple operation to reduce the number of channels into $1/k$, which makes the number of channels concatenated to the previous stage to be small for saving computation and parameter size.

**The DR-block.** The DR-block at the head is similar to the UR-block. There are only two different implementations between them. First, we use $2 \times 2$ max-pooling for down-sampling in the DR-block. Second, in the DR-block, the channel reduction function in the UR-block is not used so that the gradient at the current stage can be directly transmitted to the parameters at the previous stage. Following the UR-block in (5)-(7), the DR block can be implemented as follows:

$$\begin{aligned} x_{s+1}^h &= down(\tilde{x}_s'^h), \\ \tilde{x}_s'^h &= \tilde{x}_s^h + \mathcal{M}(\tilde{x}_s^h), \\ \tilde{x}_s^h &= concat(x_s^h, \mathcal{T}(x_s^b)), \end{aligned} \tag{9}$$

where the $x_{s+1}^h$ denotes the features at the head part for the stage $s + 1$. In this way, the features from every stage of the whole network is able to be directly connected to the final layer through concatenation, skip connection, and max-pooling. Note that we do not apply the channel-wise summation operation $r(\cdot)$ defined in (6) to obtain $\tilde{x}_s^h$ from $x_s^h$ for the DR-block in (9). Therefore, the layers obtaining $\tilde{x}_s^h$ from $x_s^h$ in the DR-block could be actually regarded as a residual block [8].

## 3.2 Detailed design and discussion

**Design of FishNet for handling the gradient propagation problem**. With the body and head designed in the FishNet, the features from all stages at the tail and body are concatenated at the head. We carefully designed the layers in the head so that there is no I-conv in it. The layers in the head are composed of concatenation, convolution with identity mapping, and max-pooling. Therefore, the gradient propagation problem from the previous backbone network in the tail are solved with the FishNet by 1) excluding I-conv at the head; and 2) using concatenation at the body and the head.

**Selection of up/down-sampling function.** The kernel size is set as $2 \times 2$ for down-sampling with stride 2 to avoid the overlapping between pixels. Ablation studies will show the effect of different kinds of kernel sizes in the network. To avoid the problem from I-conv, the weighted de-convolution in up-sampling method should be avoided. For simplicity, we choose nearest neighbor interpolation for up-sampling. Since the up-sampling operation will dilute input features with lower resolution, we apply dilated convolution in the refining blocks.

**Bridge module between the fish body and tail.** As the tail part will down sample the features into resolution $1 \times 1$, these $1 \times 1$ features need to be upsampled to $7 \times 7$. We apply a SE-block [11] here to map the feature from $1 \times 1$ into $7 \times 7$ using a channel-wise attentive operation.

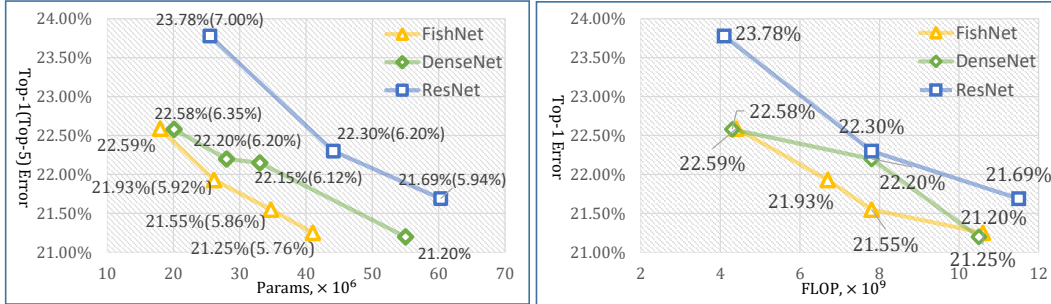

Figure 4: The comparison of the classification top-1 (top-5) error rates as a function of the number of parameters (left) and FLOP (right) for FishNet, DenseNet and ResNet (single-crop testing) on the validation set of ImageNet.

| Method | Params | Top-1 Error |
|---|---|---|
| ResNeXt-50 (32 × 4*d*) | 25.0M | 22.2% |
| FishNeXt-150 (4d) | 26.2M | **21.5%** |

Table 1: ImageNet-1k val Top-1 error for the ResNeXt-based architectures. The 4d here for FishNeXt-150 (4d) indicates that the minimum number of channels for a single group is 4.

| Method | Params | Top-1 Error |
|---|---|---|
| Max-Pooling (3 × 3, stride=2) | 26.4M | 22.51% |
| Max-Pooling (2 × 2, stride=2) | 26.4M | **21.93%** |
| Avg-Pooling (2 × 2, stride=2) | 26.4M | 22.86% |
| Convolution (stride=2) | 30.2M | 22.75% |

Table 2: ImageNet-1k val Top-1 error for different down-sampling methods based on FishNet-150.

## 4 Experiments and Results

### 4.1 Implementation details on image classification

For image classification, we evaluate our network on the ImageNet 2012 classification dataset [25] that consists of 1000 classes. This dataset has 1.2 million images for training, and 50,000 images for validation (denoted by ImageNet-1k val). We implement the FishNet based on the prevalent deep learning framework PyTorch [23]. For training, we randomly crop the images into the resolution of $224 \times 224$ with batch size 256, and choose stochastic gradient descent (SGD) as the training optimizer with the base learning rate set to 0.1. The weight decay and momentum are $10^{-4}$ and 0.9 respectively. We train the network for 100 epochs, and the learning rate is decreased by 10 times every 30 epochs. The normalization process is done by first converting the value of each pixel into the interval $[0, 1]$, and then subtracting the mean and dividing the variance for each channel of the RGB respectively. We follow the way of augmentation (random crop, horizontal flip and standard color augmentation [17]) used in [9] for fair comparison. All the experiments in this paper are evaluated through single-crop validation process on the validation dataset of ImageNet-1k. Specifically, an image region of size $224 \times 224$ is cropped from the center of an input image with its shorter side being resized to 256.This $224 \times 224$ image region is the input of the network.

FishNet is a framework. It does not specify the building block. For the experimental results in this paper, FishNet uses the Residual block with identity mapping [8] as the basic building block, FishNeXt uses the Residual block with identity mapping and grouping [29] as the building block.

### 4.2 Experimental results on ImageNet

Figure 4 shows the top-1 error for ResNet, DenseNet, and FishNet as a function of the number of parameters on the validation dataset of ImageNet-1k. When our network uses pre-activation ResNet as the tail part of the FishNet, the FishNet performs better than ResNet and DenseNet.

**FishNet vs. ResNet.** For fair comparison, we re-implement the ResNet and report the result of ResNet-50 and ResNet-101 in Figure 4. Our reported single-crop result for ResNet-50 and ResNet-101 with identity mapping is higher than that in [9] as we select the residual block with pre-activation to be our basic building block. Compared to ResNet, FishNet achieves a remarkable reduction in error rate. The FishNet-150 (21.93%, 26.4M), for which the number of parameters is close to ResNet-50

| | Instance Segmentation | Object Detection | |
| | Mask R-CNN | Mask R-CNN | FPN |
|---|---|---|---|
| Backbone | $AP^s/AP_S^s/AP_M^s/AP_L^s$ | $AP^d/AP_S^d/AP_M^d/AP_L^d$ | $AP^d/AP_S^d/AP_M^d/AP_L^d$ |
| ResNet-50 [5] | 34.5/15.6/37.1/52.1 | 38.6/22.2/41.5/50.8 | 37.9/21.5/41.1/49.9 |
| ResNet-50[†] | 34.7/18.5/37.4/47.7 | 38.7/22.3/42.0/51.2 | 38.0/21.4/41.6/50.1 |
| ResNeXt-50 (32x4d)[†] | 35.7/19.1/38.5/48.5 | 40.0/23.1/43.0/52.8 | 39.3/23.2/42.3/51.7 |
| FishNet-150 | **37.0**/19.8/40.2/50.3 | **41.5**/24.1/44.9/55.0 | **40.6**/23.3/43.9/53.7 |
| vs. ResNet-50[†] | **+2.3/+1.3/+2.8/+2.6** | **+2.8/+1.8/+2.9/+3.8** | **+2.6/+1.9/+2.3/+3.6** |
| vs. ResNeXt-50[†] | **+1.3/+0.7/+1.7/+1.8** | **+1.5/+1.0/+1.9/+2.2** | **+1.3/+0.1/+1.6/+2.0** |

Table 3: MS COCO *val-2017* detection and segmentation Average Precision (%) for different methods. $AP_*^s$ and $AP_*^d$ denote the average precision for segmentation and detection respectively. $AP_S^*$, $AP_M^*$, and $AP_L^*$ respectively denote the AP for the small, medium and large objects. The back-bone networks are used for two different segmentation and detection approaches, i.e. Mask R-CNN [10] and FPN [21]. The model re-implemented by us is denoted by a symbol [†]. FishNet-150 does not use grouping, and the number of parameters for FishNet-150 is close to that of ResNet-50 and ResNeXt-50.

(23.78%, 25.5M), is able to surpass the performance of ResNet-101 (22.30%, 44.5M). In terms of FLOPs, as shown in the right figure of Figure 4, the FishNet is also able to achieve better performance with lower FLOPs compared with the ResNet.

**FishNet vs. DenseNet.** DenseNet iteratively aggregates the features with the same resolution by concatenation and then reduce the dimension between each dense-block by a transition layer. According to the results in Figure 4, DenseNet is able to surpass the accuracy of ResNet using fewer parameters. Since FishNet preserves features with more diversity and better handles the gradient propagation problem, FishNet is able to achieve better performance than DenseNet with fewer parameters. Besides, the memory cost of the FishNet is also lower than the DenseNet. Take the FishNet-150 as an example, when the batch size on a single GPU is 32, the memory cost of FishNet-150 is 6505M, which is 2764M smaller than the the cost of DenseNet-161 (9269M).

**FishNeXt vs. ResNeXt** The architecture of FishNet could be combined with other kinds of designs, e.g., the channel-wise grouping adopted by ResNeXt. We follow the criterion that the number of channels in a group for each block (UR/DR block and transfer block) of the same stage should be the same. The width of a single group will be doubled once the stage index increase by 1. In this way, the ResNet-based FishNet could be constructed into a ResNeXt-based network, namely FishNeXt. We construct a compact model FishNeXt-150 with 26 million of parameters. The number of parameters for FishNeXt-150 is close to ResNeXt-50. From Table 1, the absolute top-1 error rate can be reduced by 0.7% when compared with the corresponding ResNeXt architecture.

### 4.3 Ablation studies

**Pooling vs. convolution with stride.** We investigated four kinds of down-sampling methods based on the network FishNet-150, including convolution, max-pooling with the kernel size of $2 \times 2$ and $3 \times 3$, and average pooling with kernel size $2 \times 2$[1]. As shown in Table 2, the performance of applying $2 \times 2$ max-pooling is better than the other methods. Stride-Convolution will hinder the loss from directly propagating the gradient to the shallow layer while pooling will not. We also find that max-pooling with kernel size $3 \times 3$ performs worse than size $2 \times 2$, as the structural information might be disturbed by the max-pooling with the $3 \times 3$ kernel, which has overlapping pooling window.

**Dilated convolution.** Yu et al. [32] found that the loss of spatial acuity may lead to the limitation of the accuracy for image classification. In FishNet, the UR-block will dilute the original low-resolution features, therefore, we adopt dilated convolution in the fish body. When the dilated kernels is used at the fish body for up-sampling, the absolute top-1 error rate is reduced by 0.13% based on FishNet-150. However, there is 0.1% absolute error rate increase if dilated convolution is used in both the fish body and head compared to the model without any dilation introduced. Besides, we replace the first $7 \times 7$ stride-convolution layer with two residual blocks, which reduces the absolute top-1 error by 0.18%.

### 4.4 Experimental investigations on MS COCO

We evaluate the generalization capability of FishNet on object detection and instance segmentation on MS COCO [20]. For fair comparison, all models implemented by ourselves use the same settings except for the network backbone. All the codes implementing the results reported in this paper about object detection and instance segmentation are released at [1].

**Dataset and Metrics** MS COCO [20] is one of the most challenging datasets for object detection and instance segmentation. There are 80 classes with bounding box annotations and pixel-wise instance mask annotations. It consists of 118k images for training (*train-2017*) and 5k images for validation (*val-2017*). We train our models on the *train-2017* and report results on the *val-2017*. We evaluate all models with the standard COCO evaluation metrics AP (averaged mean Average Precision over different IoU thresholds) [10], and the $AP_S$, $AP_M$, $AP_L$ (AP at different scales).

**Implementation Details** We re-implement the Feature Pyramid Networks (FPN) and Mask R-CNN based on PyTorch [23], and report the re-implemented results in Table 3. Our re-implemented results are close to the results reported in Detectron[5]. With FishNet, we trained all networks on 16 GPUs with batch size 16 (one per GPU) for 32 epochs. SGD is used as the training optimizer with a learning rate 0.02, which is decreased by 10 at the 20 epoch and 28 epoch. As the mini-batch size is small, the batch-normalization layers [14] in our network are all fixed during the whole training process. A warming-up training process [6] is applied for 1 epoch and the gradients are clipped below a maximum hyper-parameter of 5.0 in the first 2 epochs to handle the huge gradients during the initial training stage. The weights of the convolution on the resolution of $224 \times 224$ are all fixed. We use a weight decay of 0.0001 and a momentum of 0.9. The networks are trained and tested in an end-to-end manner. All other hyper-parameters used in experiments follow those in [5].

**Object Detection Results Based on FPN.** We report the results of detection using FPN with FishNet-150 on *val-2017* for comparison. The top-down pathway and lateral connections in FPN are attached to the fish head. As shown in Table 3, the FishNet-150 obtains a 2.6% absolute AP increase to ResNet-50, and a 1.3% absolute AP increase to ResNeXt-50.

**Instance Segmentation and Object Detection Results Based on Mask R-CNN.** Similar to the method adopted in FPN, we also plug FishNet into Mask R-CNN for simultaneous segmentation and detection. As shown in Table 3, for the task of instance segmentation, 2.3% and 1.3% absolute AP gains are achieved compared to the ResNet-50 and ResNeXt-50. Moreover, when the network is trained in such multi-task fashion, the performance of object detection could be even better. With the FishNet plugged into the Mask R-CNN, 2.8% and 1.5% improvement in absolute AP have been observed compared to the ResNet-50 and ResNeXt-50 respectively.

Note that FishNet-150 does NOT use channel-wise grouping, and the number of parameters for FishNet-150 is close to that of ResNet-50 and ResNeXt-50. When compared with ResNeXt-50, FishNet-150 only reduces absolute error rate by 0.2% for image classification, while it improves the absolute AP by 1.3% and 1.5% respectively for object detection and instance segmentation. This shows that the FishNet provides features that are more effective for the region-level task of object detection and the pixel-level task of segmentation.

**COCO Detection Challenge 2018.** FishNet was used as one of the network backbones of the winning entry. By embedding the FishNet into our framework, the single model FishNeXt-229 could finally achieve 43.3% on the task of instance segmentation on the *test-dev* set.

## 5 Conclusion

In this paper, we propose a novel CNN architecture to unify the advantages of architectures designed for the tasks recognizing objects on different levels. The design of feature preservation and refinement not only helps to handle the problem of direct gradient propagation, but also is friendly to pixel-level and region-level tasks. Experimental results have demonstrated and validated the improvement of our network. For future works, we will investigate more detailed settings of our network, e.g., the number of channels/blocks for each stage, and also the integration with other network architectures. The performance for larger models on both datasets will also be reported.

**Acknowledgement** We would like to thank Guo Lu and Olly Styles for their careful proofreading. We also appreciate Mr. Hui Zhou at SenseTime Research for his broad network that could incredibly organize the authors of this paper together.

## Footnotes

[1] When convolution with a stride of 2 is used, it is used for both the tail and the head of the FishNet. When pooling is used, we still put a $1 \times 1$ convolution on the skip connection of the last residual blocks for each stage at the tail to change the number of channels between two stages, but we do not use such convolution at the head.

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
