[Reviews · NeurIPS 2018]

Reviewer 1



This paper used a new framework called FishNet that utilizes features of different granularity (pixel-level, region-level, and image-level) in order to do the final learning task better! The motivation behind this work is that models such as CNN use consecutive down-sampling to obtain the features that are of low-resolution and miss high-resolution features that are critical for pixel-level or region-level tasks! The authors build their framework in a fish like structure where there is a tail, a body, and a head representing different parts of the model. The tail can be an existing model such as ResNet and it is in the body and head of their model where they utilize up-sampling and down-sampling to combine the features with different granularity! The authors shows through extensive experimental study that their framework works better than the baselines for different task including image classification, object detection, and instance segmentation.

Reviewer 2



-The paper describes a network structure for handling images with multiple different levels from image-level to pixel-level within a single architecture. Key insight in the work is to provide a versatile architecture for several vision tasks, not only an image classification task but also pixel-level segmentation and region-level detection tasks. This can be done by proposing a fish-like encoder-decoder-encoder structure which performs better than existing popular architectures for the aforementioned applications. The paper is well written and presents sufficient applications of this. -When seeing the architecture, it seems like an encoder-decoder-encoder structure, so one can roughly understand it as autoencoder + CNN with additional constraints. So this reviewer is wondering if you provide additional ablation study of the proposed network by removing some connections/concatenations among tail, body, head, or compared to the above mentioned structure as a baseline even though the body part does not reconstruct the original size image. By the way, is there any reason why the body part does not decode the layer size until the input size? -There exist related work with similar analysis for the third advantage about different level features described in the second page in that different level features give different abstraction of the image, complement each other, and can be combined for more accurate performance. Kim et al., “NestedNet: Learning nested sparse structures in deep neural networks”, IEEE CVPR, 2018. -In Section 3.1, the notations in the equations are somewhat complex and confusing, so I spent lots of time to understand how it works in the blocks. If you provide more generous explanation on this (+ corresponding images are much better), the revised one will be more complete and helpful for many practitioners in the community. -Could you provide details (+ rule how to construct, like ResNet) how to define the number X in FishNet-X? It is mentioned FishNet-188 in the paper (203 in the supplementary material) but there are also different FishNet-X in Figure 3.

Reviewer 3



The paper proposes a new backbone architecture more suited to various tasks, e.g. image-, region- and pixel-level tasks, than traditional pre-trained backbones. It allows direct backpropagation from deep layers to shallow layers by solving the issue caused by isolated convolutions happening in common architectures, e.g. ResNet or DenseNet. Experimental validation confirms that FishNet backbone is better than comparable ResNet and DenseNet backbones on image classification, and on both object detection and instance segmentation. The paper is rather clear and easy to follow. The motivations are exposed well and the justification of the idea to solve the issue of gradient propagation in common backbones is appealing. However, the mathematical notations are cumbersome, and not consistent across equations, e.g. the superscripts 'u' disappear between equations (4) and (5). They are also different notations between the equations and Figure 2, see k and K, or different fonts for M function. It could also help to write the name of the variables directly in Figure 2, making the link between the two easier to follow. Regarding the experimental section, the results presented are convincing on the usefulness of FishNet, especially for object detection and segmentation, where the use of recent frameworks (FPN, Mask R-CNN) is welcome. It would be interesting to have comparisons with deeper versions of ResNe(X)t backbone on MS COCO dataset, as the 101-layer versions are the most common used. It would also be good to evaluate FishNet on the test sets for easier comparisons with the literature, and to have the results from other state-of-the-art methods in the same tables, for both classification on ImageNet and object detection/segmentation on MS COCO. The ablation study seems a bit light, and does not show the relative importance of the main parts introduced, i.e. the fish body and head. ************* Post-rebuttal recommendation The rebuttal is a bit disappointing since the authors only answer to one of my concern. This said, I like the proposed method and the experiments are overall convincing. I thus stick on my weak accept recommendation.